# A Novel Approach: Investigating the Intracellular Clearance Mechanism of Glyceraldehyde-Derived Advanced Glycation End-Products Using the Artificial Checkpoint Kinase 1 d270KD Mutant as a Substrate Model

**DOI:** 10.3390/cells12242838

**Published:** 2023-12-14

**Authors:** Kenji Takeda, Akiko Sakai-Sakasai, Kouji Kajinami, Masayoshi Takeuchi

**Affiliations:** 1Department of Advanced Medicine, Medical Research Institute, Kanazawa Medical University, 1-1 Daigaku, Uchinada-Machi, Ishikawa 920-0293, Japan; asakasai@kanazawa-med.ac.jp (A.S.-S.); takeuchi@kanazawa-med.ac.jp (M.T.); 2Department of Cardiology, Kanazawa Medical University, 1-1 Daigaku, Uchinada-Machi, Ishikawa 920-0293, Japan; kajinami@kanazawa-med.ac.jp

**Keywords:** advanced glycation end-products (AGEs), glyceraldehyde (GA), glyceraldehyde-derived AGEs, toxic AGEs (TAGEs), p62/SQSTM1, CHK1-CPs

## Abstract

Advanced glycation end-products (AGEs), formed through glyceraldehyde (GA) as an intermediate in non-enzymatic reactions with intracellular proteins, are cytotoxic and have been implicated in the pathogenesis of various diseases. Despite their significance, the mechanisms underlying the degradation of GA-derived AGEs (GA-AGEs) remain unclear. In the present study, we found that N-terminal checkpoint kinase 1 cleavage products (CHK1-CPs) and their mimic protein, d270WT, were degraded intracellularly post-GA exposure. Notably, a kinase-dead d270WT variant (d270KD) underwent rapid GA-induced degradation, primarily via the ubiquitin–proteasome pathway. The high-molecular-weight complexes formed by the GA stimulation of d270KD were abundant in the RIPA-insoluble fraction, which also contained high levels of GA-AGEs. Immunoprecipitation experiments indicated that the high-molecular-weight complexes of d270KD were modified by GA-AGEs and that p62/SQSTM1 was one of its components. The knockdown of p62 or treatment with chloroquine reduced the amount of high-molecular-weight complexes in the RIPA-insoluble fraction, indicating its involvement in the formation of GA-AGE aggregates. The present results suggest that the ubiquitin–proteasome pathway and p62 play a role in the degradation and aggregation of intracellular GA-AGEs. This study provides novel insights into the mechanisms underlying GA-AGE metabolism and may lead to the development of novel therapeutic strategies for diseases associated with the accumulation of GA-AGEs.

## 1. Introduction

Sugars are a vital source of energy and essential nutrients for living organisms. However, modifications to the lysine and arginine residues of proteins in vivo, which result in the formation of crosslinks, may significantly change the steric structures of proteins, thereby affecting their activities and physical properties. This process is known as glycation or the Maillard reaction and is divided into two stages: an early-stage reaction that produces Amadori transfer products and a late-stage reaction that results in advanced glycation end-products (AGEs) through oxidation, dehydration, and condensation [1]. AGEs have been implicated in the pathogenesis of various diseases [2,3], such as diabetes, liver disease [4], atherosclerosis [5], Alzheimer’s disease, and aging [6].

AGEs represent a diverse cluster of compounds. For instance, Nε-(carboxymethyl)lysine (CML) and Nε-(carboxyethyl)lysine (CEL) are AGEs formed through reactions with glyoxal (GO), methylglyoxal (MGO), and lysine residues in proteins. AGEs can be broadly classified into exogenously derived dietary AGEs, formed through the heat processing of foods like milk and dairy products [7,8], and endogenously generated AGEs that accumulate within the body. The development of diseases due to endogenous AGE accumulation is postulated to involve two pathways: (1) the activation of downstream cellular signaling through the RAGE receptor, inducing oxidative stress and inflammation [9] and (2) cross-linking with intracellular proteins, disrupting normal physiological functions. Additionally, reports emphasize the AGE interactions with the gut microbiota and their consequential effects [7].

AGEs are generated in vivo not only from glucose, but also from, for example, metabolic intermediates of glucose, degradation products, and Maillard reaction intermediates [10,11]. In the classification spectrum of more than 40 AGEs identified to date [9], glyceraldehyde (GA) plays a crucial role in the formation of GA-AGEs (GA-derived AGEs) through intricate metabolic pathways [3]. In fructolysis, fructokinase (FK) phosphorylates fructose (Fru), forming Fru-1-phosphate (F-1-P). The subsequent cleavage by aldolase B yields dihydroxyacetone-phosphate and GA. Concurrently, glycolysis metabolizes GA-3-phosphate (GA-3-P) through GA-3-P dehydrogenase (GAPDH), ultimately leading to pyruvate. Reduced GAPDH activity results in intracellular accumulation of GA-3-P, promoting non-enzymatic reactions that contribute to GA formation. Moreover, under hyperglycemic conditions, the polyol pathway is activated and glucose is reduced by aldose reductase to form sorbitol, which is subsequently oxidized by sorbitol dehydrogenase to form Fru. These excess GAs, arising from impaired glucose metabolism, non-enzymatically react with proteins, ultimately leading to the formation of the GA-AGEs implicated in diseases associated with diabetic complications, insulin resistance, heart disease, Alzheimer’s disease, hypertension, nonalcoholic steatohepatitis, and cancer [3]. The toxic effects of GA-AGEs and their specific structures that are responsible for toxicity remain unclear. However, an antibody targeting GA-AGEs was shown to effectively mitigate neurotoxicity caused by serum AGEs in patients with diabetic nephropathy undergoing hemodialysis [12]. Other antibodies targeting different types of AGEs or CML did not exert similar protective effects. These findings indicate that only AGE structures containing epitopes recognized by the anti-GA-AGE antibody are toxic. They are referred to as toxic AGEs (TAGEs) and are distinct from other known GA-AGEs, such as 3-hydroxy-5-hydroxymethyl-pyridinium (GLAP), triosidines, and MG-H1. None of these structures showed AGE-specific fluorescence or protein cross-linking. Two compounds with a 1,4-dihydropyrazine ring that showed fluorescence and had cross-links were identified as TAGE candidate structures (PCT/JP2019/34195).

The late-stage Maillard reaction is irreversible and once GA-AGEs are formed, there are no known enzymes that specifically remove the sites modified by glycation. Glycation modifications have been suggested to compromise cellular homeostasis by inducing the loss of function of various biomolecules or by forming toxic aggregates, which inactivate other normal essential proteins [3]. The ubiquitin–proteasome pathway and autophagy pathway are intracellular systems that remove defective proteins and aggregates. These two pathways are not independent of each other, and they function cooperatively [13,14,15,16]. Recent studies suggested that AGEs are closely associated with autophagy [17]. Their effects on autophagy are complex, with some studies indicating that they inhibit autophagy [18,19] and others suggesting that they induce autophagy [20,21,22,23,24]. Therefore, the coordinated role of the two intracellular degradation mechanisms in the AGE clearance process remains unclear. Although AGEs are formed and accumulate in proteins with a long turnover due to a decrease in the activities of intracellular protein quality control mechanisms with aging, the mechanisms underlying intracellular GA-AGE degradation and removal have yet to be clarified. A more detailed understanding of these mechanisms is critical for the development of effective therapeutic strategies that mitigate the detrimental effects of GA-AGE accumulation.

In the present study, we found that N-terminal checkpoint kinase 1 cleaved products (CHK1-CPs) and their mimetic protein (d270WT) were susceptible to intracellular degradation upon the administration of GA, and also that a kinase-dead mutant (d270KD) of d270WT exhibited more rapid GA-responsive degradation with the formation of high-molecular-weight complexes (typical features indicative of GA-AGE conversion). d270KD, which is rapidly degraded with the generation of GA-AGEs, may serve as a useful model for GA-AGE formation in future research on the molecular mechanisms that act on the clearance of GA-AGEs that accumulate in cells.

## 2. Materials and Methods

### 2.1. Antibodies

The following primary antibodies were used: anti-mouse IgG2a magnetic beads (M076-11), anti-DYKDDDDK (FLAG)-tag magnetic beads (M185-11), anti-V5-tag magnetic beads (M16711), anti-normal rabbit IgG (PM035), and an anti-DYKDDDDK (FLAG)-tag (PM020) for immunoprecipitation, and anti-HA-tag (M132-3), anti-α-tubulin-HRP, and anti-microtubule-associated protein 1 light chain 3 (LC3; M186-3) for Western blotting from Medical & Biological Laboratories (MBL, Nagoya, Japan); anti-phospho-SQSTM1/p62 (Ser403) (#39786), anti-SQSTM1/p62 (#5114), and anti-HUWE1 (#5695) from Cell Signaling Technologies (CST, Beverley, MA, USA); anti-V5-HRP (R961-25) from Thermo Fisher Scientific (Waltham, MA, USA); anti-FLAG-HRP (015-22391) from Wako Pure Chemicals (Osaka, Japan); anti-ubiquitin (#AUB01) from Cytoskeleton (Denver, CO, USA); and anti-SPRTN (HPA025073) from Sigma-Aldrich (St Louis, MO, USA). An anti-GA-AGE (TAGE) antibody was prepared as previously reported [25]. The anti-TAGE antibody specifically identified unique epitopes other than the known structures derived from GA, such as GLAP and triosidines. However, it did not recognize well-known AGEs containing CML and Nε-(carboxyethyl)lysine (CEL) or bind to other AGEs derived from reducing sugars and carbonyl molecules, including pyrraline, pentosidine, crossline, argpyrimidine, GO- or MGO-lysine dimers, and GO- or MGO-derived hydroimidazolone.

### 2.2. Cell Culture

COS-7 cells were obtained from the American Type Culture Collection and cultured in Dulbecco’s modified Eagle medium (DMEM + GlutaMax medium; Gibco BRL, Gland Island, NY, USA) supplemented with 10% fetal bovine serum (FBS; JRH Biosciences, Lenexa, KS, USA), 100 U/mL penicillin, and 100 µg/mL streptomycin (both from Gibco BRL) at 37 °C in a humidified atmosphere of 5% CO_2_. HeLa cells (a gift from Dr. Sumiyo Akazawa, Kanazawa Medical University, Japan) were cultured in minimum essential medium (MEM alpha + GlutaMax medium; Gibco BRL, Waltham, MA, USA) supplemented with 10% FBS, 100 U/mL penicillin, and 100 µg/mL streptomycin at 37 °C in a humidified atmosphere of 5% CO_2_. Cells were subcultured when they reached 70–80% confluence. In experiments where GA (purity 98% or higher; catalog number: 17014-94, Nacalai Tesque, Tokyo, Japan) was applied to cells, GA or phosphate buffer was added at the concentrations indicated. In the aminoguanidine (AG; purity 97% or higher; catalog number: 328-26432, FUJIFILM Wako Pure Chemical Corporation, Osaka, Japan)-induced suppression experiment, AG at the indicated concentrations was added to cells 2 h prior to the treatment with GA. Regarding the proteasome inhibitor treatment, cells were treated with 10 µM MG-132 (Sigma-Aldrich, St. Louis, MO, USA) or DMSO (Nacalai Tesque, Kyoto, Japan) for 6 h. In the autophagy inhibition experiments, 50 µM chloroquine (Sigma-Aldrich) or phosphate buffer was added to HeLa cells 20 h prior to the GA treatment.

### 2.3. Lactate Dehydrogenase (LDH) Cytotoxicity Assay

LDH released into the culture supernatant was measured using a LDH cytotoxicity assay kit (Nacalai Tesque, Kyoto, Japan) following the manufacturer’s instructions. Briefly, HeLa cells were plated on 24-well plates (5  × 10^4^ cells/well). The cells were stimulated with 4 mM GA or phosphate buffer for the indicated time. The culture medium (100 μL) was collected and mixed with 100 μL of the substrate solution. After an incubation at room temperature for 20 min, 50 μL of the stop solution was added and the absorbance was measured at 490 nm using an iMark™ Microplate Reader (Bio-Rad, Hercules, CA, USA). LDH levels in the cell culture supernatant collected from the GA-treated cell group were normalized relative to the cultured medium collected from the control group (phosphate buffer-treated group).

### 2.4. Generation of Expression Plasmid Constructs

Flag-tagged CHK1 expression constructs, including the wild-type (WT) form and its kinase-dead mutant form (KDmut), were previously described [26]. Briefly, cDNA encoding full-length rat Chk1 was amplified from rat cardiac myocyte cDNA using a forward primer containing the Flag-tag sequence at the 5′ end and then subcloned into the pcDNA4/HisMax vector. Xpress- and 6 × His-tag sequences at the N terminus were removed from the vector by PCR using the KOD-plus-mutagenesis kit (Toyobo, Osaka, Japan) according to the manufacturer’s instructions. Deletion at the C terminus and point mutations in Flag-Chk1 expression vectors were created by PCR using the KOD-plus-mutagenesis kit. To generate the d270KD-EGFP-V5 expression construct, full-length EGFP cDNA was amplified by PCR using the pEGFP-C1 vector (BD Biosciences, San Jose, CA, USA) and subcloned into the pcDNA3.1 vector (Thermo Fisher Scientific). The open reading frame of EGFP with the V5-tag and 6 × His-tag sequences at the C terminus was then amplified using the vector as a template by PCR using PrimeSTAR Max DNA polymerase, subcloned into the pENTR/D-TOPO vector (Thermo Fisher Scientific) using the TOPO^®^ cloning procedure, and then transformed into the mammalian expression vector pDEST30 (Thermo Fisher Scientific) using LR clonase II enzyme. Fragments containing the N-terminal region of Chk1 (amino acids 1-270) in pFlag-Chk1Wt and its KDmut (d270KD) vectors were cloned in the vector in-frame with the gene that encodes the EGFP-V5/6 × His-tag at the 3′ end using the In-Fusion^®^ HD PCR cloning kit (Takara, Shiga, Japan). To generate the d270KD-ZsGreen expression construct, full-length ZsGreen1 cDNA was amplified by PCR using the ZsGreen1-1 vector (Clontech, Palo Alto, CA, USA) and the fragments were cloned into the vector in-frame with the gene encoding Flag-d270KD at the 3′ end using the In-Fusion^®^ HD PCR cloning kit (Takara, Shiga, Japan). HA-ubiquitin was a gift from Edward Yeh (Addgene plasmid #18712). All constructs were sequenced to ensure proper ligation in the correct frame and Taq polymerase fidelity using the ABI PRISM TM 310 genetic analyzer (Applied Biosystems, Foster City, CA, USA).

### 2.5. Transfection

Regarding transient plasmid DNA transfection, cells were seeded on 35, 60, 100 mm, or 6-well tissue culture plates, cultured in complete growth medium, and then transfected using FuGENE^®^ 4K (Promega, Madison, WI, USA) according to the manufacturer’s protocol. The transfection of siRNA was performed using the RNAiMax Transfection reagent (Thermo Fisher Scientific) according to the manufacturer’s recommendations. SignalSilence^®^ siRNA (#6241, Cell Signaling Technologies) targeting Chk1, Silencer Select siRNA (Ambion, Valencia, CA, USA) targeting Huwe1 (siRNA ID: s19596 and s19597), SQSTM1 (siRNA ID: s16962), SPRTN (siRNA ID: s38329), or scrambled siRNA (Silencer Select Negative Control #1, catalogue #4390843) was used. The final siRNA concentration was 50 nM for SignalSilence^®^ siRNAs and 5 nM for Silencer Select siRNAs.

### 2.6. Western Blot

Transfected COS-7 and HeLa cells were lysed in lysis buffer (CelLytic™M cell lysis reagent; Sigma-Aldrich) containing proteinase inhibitors and phosphatase inhibitors (both from Nacalai Tesque). After the cellular debris was removed by centrifugation, protein concentrations in the supernatants were measured using the Qubit protein assay kit (Thermo Fisher Scientific). To detect GA-AGEs (TAGE) or high-molecular-weight complexes, cells were lysed in RIPA buffer (50 mM Tris-HCl pH 7.5, 150 mM NaCl, 1% NP-40, 0.5% Na deoxycholate, and 0.1% SDS), centrifuged, and separated into supernatant (soluble fraction) and pellet (insoluble fraction). After the addition of SDS sample buffer to each fraction, the RIPA-insoluble fraction was sonicated. All samples were boiled at 95 °C for 5 min, separated on 5–20 or 12.5% SDS-PAGE gels, and analyzed by Western blotting. Target proteins were visualized by Chemi-Lumi One L, Super (Nacalai Tesque), or Immobilon Forte Western HRP Substrate (Millipore, Bedford, MA, USA). A densitometric quantification of the resultant blots was performed using NIH ImageJ software (version 1.54d). 

### 2.7. In Vitro GA-AGE Modification Assay

The in vitro GA-AGE modification of the d270KD protein was performed according to previous methods with some changes. Briefly, HeLa cells transfected with Flag-tag-fused d270KD (Flag-d270KD) were lysed in CelLytic™M cell lysis reagent containing proteinase inhibitors and phosphatase inhibitors. After the cellular debris was removed by centrifugation, total cellular proteins were incubated with anti-Flag antibody-immobilized magnetic beads at 4 °C overnight, and the resulting immunoprecipitates were washed three times with lysis reagent. After the elution of Flag-d270KD with 40 µL of Flag peptide (2 mg/mL), 200 µL of PBS was added. The eluted samples, including Flag-d270KD, were then ultrafiltered and concentrated with Amicon Ultra 0.5 (10 K) (Millipore) to remove the FLAG peptide. This purified Flag-d270KD recombinant protein (2.5 µg) was incubated in 50 µL of PBS for 20 h with or without the addition of 4 mM GA. After the addition of SDS sample buffer to these reaction mixtures, they were boiled at 95 °C for 5 min. 

### 2.8. In Vivo Ubiquitination Assay

For the in vivo ubiquitylation assays, the d270KD-EGFP expression vector d270KD-EGFP was used to transfect COS-7 cells (100 mm plates) for 48 h. The cells were treated with 10 µM MG132 and then harvested after 8 h. The cell pellets were resuspended in denaturing buffer (1.5% SDS, 50 mM Tris–HCl pH 7.5, and 5 mM DTT) followed by boiling for 10 min, and were then diluted 10-fold with CelLytic™M cell lysis reagent containing proteinase inhibitors and phosphatase inhibitors. After the cellular debris was removed by centrifugation, the extracted proteins were immunoprecipitated with anti-V5 antibody-immobilized magnetic beads and subjected to the assay described above. Immunoprecipitates were analyzed by Western blotting with anti-ubiquitin antibodies or anti-V5 antibodies. 

### 2.9. Luciferase Assay

The gene encoding luciferase was amplified by PCR using the pGL4.54 vector (Promega) as a template with the forward primer 5′-ACAAACACACTTAACATGGAAGATGCCAAAAAC-3′ and reverse primer 5′-GTAACAGGCCTTCTACACGGGCGATCTTGCCGCCC-3′. The pFlag-d270KD plasmid vector was amplified by PCR using the forward primer 5′-TA GAAGGGCCTGTACCTAGGATCCAGT-3′ and reverse primer 5′-GTTAAGTGGTTTGTTATACCATCTA-3′ to form a linear strand. The luciferase gene was then inserted downstream of the d270KD gene using the InFusion HD cloning kit (Clontech, Palo Alto, CA, USA). The accuracy of the gene insertion site was confirmed by a sequence analysis. Twenty-four hours after the transfection of the d270KD fusion luciferase expression vector (d270KD-Luc) into cells cultured in 6-well plates, the firefly luciferase reporter assay was performed using the Luciferase Reporter Assay System (Promega Madison, WI, USA). Three independent assays were performed under each condition.

### 2.10. Fluorescence Imaging Analysis

COS-7 cells were grown in six-well plates containing collagen-coated glass coverslips (diameter of 12 mm, Iwaki, Shizuoka, Japan) and transfected with the d270KD-EGFP expression vector at 60% confluence using FuGENE^®^ 4K as described above. After an incubation for 24 h in complete medium, the cells were treated with or without GA (2 mM) for the indicated time and fixed with ice-cold 4% paraformaldehyde in PBS for 10 min. The fixed cells were washed three times for 5 min each with PBS and then mounted with Prolong gold antifade reagent/DAPI (Thermo Fisher Scientific). All fluorescence images were obtained using a digital high-definition microscope system (BZ-9000, Keyence, Osaka, Japan) with the following filter sets: OP-66834, Ex360/40 Em460/50; OP-66836, Ex470/40 Em535/50; OP-66838, Ex560/40 Em630/60.

### 2.11. Immunoprecipitation Assay of High-Molecular-Weight Complexes Formed by a GA Stimulation

The d270KD-ZsGreen expression vector was transfected into HeLa cells (100 mm plates) for 48 h. The cells were treated with 2 mM GA and then harvested after 3 h. The cells were lysed in RIPA buffer containing proteinase inhibitors and phosphatase inhibitors, centrifuged, and separated into supernatant (soluble fraction) and pellet (insoluble fraction). After the addition of solubilization buffer containing a high concentration of SDS (2.0% SDS, 50 mM Tris-HCl pH 7.5) to the pellet, the RIPA-insoluble fraction was sonicated. After removing the high concentration of SDS from the fraction using the Pierce™ SDS-PAGE Sample Prep Kit (Thermo Fisher Scientific), ultrafiltration was performed using Amicon Ultra 0.5 (30 K) (Millipore), while the solvent of the fraction was replaced with CelLytic™M cell lysis reagent containing proteinase inhibitors and phosphatase inhibitors. The anti-FLAG antibody (MBL) or normal rabbit IgG (MBL) was added to the soluble and insoluble fractions, which were then incubated at 4 °C overnight. The coupling of the antibodies to Dynabeads Protein A (Thermo Fisher Scientific) was performed by adding CelLytic™M-equilibrated Dynabeads directly to both fractions, followed by an incubation at room temperature for 30 min. Immunoprecipitated Dynabead complexes were washed once with CelLytic™M reagent and three times with TBS-T. Proteins bound to the Dynabead–antibody complexes were eluted by adding 50 µL of the FLAG peptide (2 mg/mL). After the addition of SDS sample buffer to each fraction, all samples were boiled at 95 °C for 5 min, separated on 5–20% SDS-PAGE gels, and analyzed by Western blotting with the indicated antibodies.

### 2.12. Statistical Analysis

All numerical results are reported as the mean ± SEM of at least three measurements. Statistical analyses were performed using the paired Student’s *t*-test, a one-way ANOVA followed by Dunnett’s post hoc test, or a two-way ANOVA followed by the Bonferroni post hoc test with GraphPad Prism software (Version 5.0, GraphPad Prism Software, San Diego, CA, USA). Differences were considered to be significant at *p* < 0.05 and were denoted by an asterisk in the graphs.

## 3. Results

### 3.1. Identification of a Constitutive KD Mutant of CHK1 That Is Rapidly Modified with GA-AGEs and Degraded in Cells after GA Stimulation

We previously reported that the prolonged stimulation of cells with GA, an intermediate metabolite of GA-AGEs, induced DNA damage stress and necrosis [3]. CHK1 is a serine–threonine kinase that plays a central role in cell cycle checkpoints and DNA stress responses [27]. However, the effects of the GA stimulation on CHK1 (GA-AGE modifications) in cells remain unclear. Therefore, we initially stimulated HeLa cells with 2 mM GA and observed the formation of high-molecular-weight complexes of endogenous CHK1 protein after 4 h (Figure 1a). Furthermore, when a FLAG-tag-fused CHK1 (Flag-CHK1) expression vector was introduced into HeLa cells and stimulated with GA in the same manner, the formation of high-molecular-weight complexes of exogenous CHK1 protein was strongly promoted (Figure 1b).

The C terminus of CHK1 was previously reported to be cleaved by SPRTN metalloprotease, resulting in N-terminal CHK1 fragments that were more active than the full-length protein [28]. In our experimental model using HeLa cells, a few N-terminal CHK1 fragments of endogenous CHK1 were detected in the basal state; however, a number of N-terminal CHK1 cleavage products (CPs) were clearly present in HeLa cells transiently expressing Flag-CHK1 (Figure 1b). As expected, the protein levels of N-terminal CHK1-CPs detected at approximately 30 kDa were suppressed in cells transfected with siRNA against SPRTN (Figure 1c). These results are consistent with previous findings showing that SPRTN cleaved the C terminus of CHK1 at multiple sites to form these N-terminal CHK1 cleaved forms. Upon a stimulation with 2 mM GA, a decrease in N-terminal CHK1-CPs was observed after 1 h. On the other hand, non-cleaved (i.e., full-length) Flag-CHK1 showed no GA-stimulated reductions in monomeric bands within 4 h. Therefore, N-terminal CHK1-CPs were more susceptible to the effects of GA-stimulated glycation modifications earlier. 

To investigate the GA-stimulated degradation of N-terminal CHK1-CPs in more detail, we generated a mutant expression vector (Flag-d270WT) lacking the C-terminal regulatory domain, as in N-terminal CHK1-CPs. As expected, a Western blot analysis showed a time-dependent decrease in the protein levels of Flag-d270WT transiently expressed in HeLa cells upon a stimulation with 2 mM GA (Figure 2a). In addition, a band of high-molecular-weight complexes of Flag-d270WT, which are characteristic of glycation modifications (GA-AGE conversion), was weakly detected from 2 h after the GA stimulation. The KD mutant (Flag-d270KD), in which the kinase was constitutively inactivated by a single amino acid substitution (a mutation in which the 36th alanine is replaced by phenylalanine: A36F) inside d270WT, showed faster degradation upon the GA stimulation than the wild type (d270WT). The majority of d270KD bands disappeared approximately 8 h after the stimulation with 2 mM GA, while d270WT maintained an expression level of approximately 40% even after 8 h of the GA stimulation. 

We then investigated the concentrations of GA that induced the rapid degradation of d270KD. The GA stimulation for 6 h resulted in significant degradation from a concentration of 1.5 mM (Figure 2b). These results indicate that d270KD was rapidly degraded upon intracellular GA-AGE modifications by a short-term GA stimulation. In contrast to d270WT, no GA-stimulated high-molecular-weight complex bands were observed in d270KD (Figure 2a), suggesting that an inactivating mutation in the wild-type form changed the structure to be less susceptible to glycation modifications by GA. We then performed GA-AGE conversion experiments on d270KD in vitro in the absence of a degradation system. Flag-d270KD was expressed in cells, and the recombinant protein was purified by immunoprecipitation with a FLAG antibody and then reacted with GA in a tube for 20 h. The results obtained showed that the GA stimulation resulted in the appearance of high-molecular-weight complexes, which were detected by anti-TAGE antibodies (Figure 2c). Therefore, the d270KD mutant was considered to be TAGE-modified by GA as well as d270WT; presumably, high-molecular-weight complexes were not detected in these cells because GA-stimulated degradation was accelerated. 

### 3.2. The Intracellular Destabilizing Property of d270KD upon GA Stimulation May Be Retained after Fusion to Other Proteins 

We examined whether the rapid degradation of d270KD induced by the GA stimulation is a property that will be retained after fusion to other proteins. d270KD-EGFP, a fusion of EGFP to the C terminus of d270KD, was expressed in COS-7 cells and then stimulated with 4 mM GA. A Western blot analysis showed that d270KD-EGFP was markedly degraded after the GA stimulation for approximately 6 h (Figure 3a).

Slightly different from the degradation pattern of Flag-d270KD, high-molecular-weight complex bands were clearly observed in the EGFP fusion 2 h after the GA stimulation. A fluorescence imaging analysis was performed using cells expressing d270KD-EGFP. The results obtained showed that when the cells were not stimulated, their fluorescence was extensively observed in the cytoplasm, whereas after the GA stimulation for 6 h, the fluorescent area decreased and only small granule-like signals were noted (Figure 3b). Furthermore, when d270KD-Luc, a luciferase fused to the C terminus of d270KD, was expressed in HeLa cells, rapid proteolysis was observed with the appearance of high-molecular-weight bands upon the GA stimulation, similar to d270KD-EGFP (Figure 4a). Consistent with the proteolysis of d270KD-Luc, its luciferase activity also decreased in concentration- and time-dependent manners upon GA stimulation (Figure 4b). We also investigated the effects of AG, a known inhibitor of AGE formation, on the GA-stimulated rapid degradation of 270KD-Luc. We found that the AG pretreatment inhibited the decrease stimulated by GA in luciferase activity in a concentration-dependent manner (Figure 4c). The Western blot analysis confirmed that this effect was due to the inhibition of the GA-induced proteolysis of d270KD-Luc (Figure 4d).

### 3.3. The GA-Stimulated Pathway by Which d270KD Undergoes Ubiquitination and Degradation Does Not Appear to Involve Mule/HUWE1 E3 Ubiquitin Ligase

In a human pancreatic beta cell line (1.4E7 cells), GA stimulation has been suggested to inhibit the autophagy pathway, increase intracellular GA-AGE levels, and induce cell death [18]. However, the effects of a GA stimulation on the ubiquitin–proteasome pathway have remained unclear. The cytotoxicity of HeLa cells induced by a 2 mM GA stimulation, as measured by the LDH assay, was approximately 5% of the control at 12 h, indicating that toxicity was still low (Appendix A). Flag-d270KD began to degrade within 2 h of the GA stimulation (Figure 2a), and presumably, intracellular GA-AGE levels were low at that time, suggesting that the degradation system consisting of the ubiquitin–proteasome pathway and autophagy pathway was functioning. The pretreatment of HeLa cells with MG132, a proteasome inhibitor, clearly produced detectable levels of endogenous N-terminal CHK1-CPs (Figure 5a), and the expression levels of these CHK1-CPs were only marginally affected by the 2 mM GA stimulation until at least 3 h (Figure 5b). These results suggest that N-terminal CHK1-CPs are subject to regulation via proteasomal degradation in the basal state of cells. 

Similarly, the pre-administration of MG132 completely blocked the degradation of d270KD by GA (Figure 6a). Additionally, stimulation with MG132 and GA promoted the formation of high-molecular-weight d270KD-EGFP complexes and increased their level of ubiquitination more than the stimulation with MG132 alone (Figure 6b). Collectively, these results suggest that the GA stimulation induced GA-AGE modifications with the rapid ubiquitination of d270KD, which was mainly degraded through the ubiquitin–proteasome pathway.

The E3 ubiquitin ligase that acts in the ubiquitination of d270KD has not yet been identified. Mule/HUWE1, a Homologous to E6-AP Carboxyl Terminus (HECT) E3 ubiquitin ligase, was previously shown to be involved in regulating the basal level of the CHK1 protein [29]. In the experimental system of HeLa cells used in the present study, when Mule/HUWE1 was knocked down with siRNA, an increase in endogenous CHK1 levels and a decrease in turnover were observed in the chase experiment with cycloheximide (Figure 7a). Similar results were obtained with the transiently expressed Flag-fused CHK1 protein (Figure 7b). We then investigated whether Mule/HUWE1 played a role in the GA-stimulated degradation of d270KD. The knockdown of Mule/HUWE1 did not suppress the degradation of d270KD (Figure 7c). It also did not affect the formation of high-molecular-weight complexes. Therefore, another ubiquitin ligase besides Mule/HUWE1 may be involved in the GA-stimulated d270KD degradation pathway. Alternatively, the GA-stimulated degradation of d270KD may involve a different pathway from that regulated at the basal level of CHK1WT because GA-stimulated d270KD undergoes GA-AGE modifications as well as ubiquitination.

### 3.4. GA-Stimulated High-Molecular-Weight Protein Complexes and GA-AGE-Modified Protein Complexes Are Resistant to Detergents

The d270KD protein expressed in HeLa cells appeared exclusively as high-molecular-weight complexes in the insoluble fraction of RIPA buffer over time after the GA stimulation (Figure 8a). Based on this result, it was assumed that not only the d270KD-derived GA-AGE-modified protein, but also many other GA-AGEs were abundant in the RIPA-insoluble fraction. As expected, a Western blot analysis of the RIPA-insoluble fractions of cells treated with or without 2 mM GA for 10 h revealed that the GA-treated samples were enriched with high-molecular-weight complexes that were strongly detected by anti-TAGE antibodies in a time-dependent manner upon GA stimulation (Figure 8b). Therefore, when intracellular proteins are modified with GA-AGEs by GA, they appear to accumulate intracellularly as aggregates that are resistant to common detergents, such as SDS.

### 3.5. p62 Is Involved in the GA-Induced Formation of High-Molecular-Weight d270KD Complexes

We hypothesized that the bulk degradation mechanism by the autophagy pathway, rather than the proteasome pathway, may function in the degradation of high-molecular-weight aggregates. Therefore, we investigated changes in the GA-stimulated behavior of p62, which plays an important role in selective autophagy by binding to ubiquitinated proteins and transporting them to autophagosomes. A single band of p62 was detected in the soluble and insoluble fractions of RIPA; in the RIPA-soluble fraction, p62 levels decreased over time upon GA stimulation, whereas a slight increase was observed in its levels in the RIPA-insoluble fraction (Figure 8a). On the other hand, the phosphorylated form of serine 403, which increases binding affinity to ubiquitinated proteins [30,31], appeared to increase in the RIPA-insoluble fraction over time upon GA stimulation. This result suggested that the GA stimulation induced the binding of p62 to GA-AGE-modified high-molecular-weight complexes that were abundant in the RIPA-insoluble fraction. 

We also examined the effects of the p62 siRNA knockdown on the GA-induced rapid degradation of d270KD. Contrary to our expectations, the p62 knockdown did not affect the degradation of d270KD in the RIPA-soluble fraction (Figure 9a), while a similar study on the RIPA-insoluble fraction showed a slight decrease in the high-molecular-weight complexes of d270KD in cells in which p62 was knocked down (Figure 9b). We then investigated whether p62 bound to the high-molecular-weight complexes of d270KD formed by the GA stimulation. To achieve this, we generated a new protein expression vector (d270KD-ZsGreen) with ZsGreen1 fused to the C terminus of d270KD and confirmed that the high-molecular-weight complexes of this fusion protein were formed within a short time after the GA stimulation and were rapidly degraded in cells (Appendix A). HeLa cells were transfected with the d270KD-ZsGreen expression vector and 48 h later, the cells were stimulated with 2 mM GA for 3 h to form the high-molecular-weight complexes of d270KD-ZsGreen. The RIPA-soluble and RIPA-insoluble fractions were then immunoprecipitated with the anti-FLAG antibody to purify the d270KD-ZsGreen complexes. As a control, the same immunoprecipitation was performed with normal rabbit IgG. 

The Western blot analysis showed that in the RIPA-soluble and insoluble fractions, p62 bands were only detected by immunoprecipitation with the anti-FLAG antibody (Figure 10). In the RIPA-insoluble fraction, not only the monomer, but also a smeared band of high-molecular-weight p62 were observed. This result indicates that GA-modified p62 was present as a component of the high-molecular-weight complex of d270KD-ZsGreen. Furthermore, the immunoprecipitation products of d270KD-ZsGreen with the FLAG antibody showed a band of high-molecular-weight complexes detected with the anti-TAGE (GA-AGE) antibody only in the RIPA-insoluble fraction. These results suggest that p62 was one of the components of the high-molecular-weight complexes of d270KD modified with GA-AGEs by the GA stimulation. 

### 3.6. The Inhibition of Autophagy Does Not Affect the Rapid Degradation of d270KD due to GA Stimulation, but Inhibits the Formation of Its High-Molecular-Weight Complexes

We investigated whether the treatment with the autophagy inhibitor chloroquine affected GA-stimulated d270KD degradation. As shown in Figure 11, after the transient expression of d270KD in HeLa cells, the pretreatment with 50 μM chloroquine did not affect GA-stimulated d270KD degradation. The administration of chloroquine increased the LC3-II/LC3-I ratio, suggesting that chloroquine inhibited the fusion of lysosomes with autophagosomes, thereby suppressing the degradation of LC3-II in the phagophore membrane. In addition, an increase in the basal level of p62 and the concomitant polymerization of p62 were observed (Figure 11). This result is consistent with previous findings showing that p62 bound to ubiquitinated proteins which were degraded together in a selective autophagy mechanism, resulting in the accumulation of p62 when autophagy was inhibited. Similar to the chloroquine-insensitive results obtained for d270KD degradation by GA, no significant changes in the expression levels of endogenous CHK1-CPs were noted upon chloroquine treatment (Appendix A).

To examine the effect of chloroquine treatment on d270KD degradation in more detail, samples were separated into RIPA-soluble and RIPA-insoluble fractions for analysis. The results showed that in the RIPA-soluble fraction, a slight increase in the basal level of d270KD was observed upon chloroquine treatment in the untreated GA fraction (Figure 12a, 0 h). However, no effect of chloroquine was observed in GA-stimulated degradation of d270KD. On the other hand, an analysis of the RIPA-insoluble fraction showed a slight decrease in the high-molecular-weight complex level of d270KD by chloroquine treatment after 8 h of GA stimulation (Figure 12b). Interestingly, p62, which becomes high-molecular-weight upon GA stimulation, accumulated at higher levels in the RIPA-insoluble fraction upon chloroquine treatment. These results suggest that the autophagy pathway is partially involved in the regulation of the basal level of d270KD, but its inhibition has little effect on GA-stimulated degradation. It is conceivable that when autophagy is inhibited by chloroquine and yet stimulated by GA, p62 abnormally accumulates in the insoluble fraction, resulting in a slight decrease in the formation of the high-molecular-weight complexes of d270KD.

## 4. Discussion

Under hyperglycemic conditions, glucose, fructose, and its metabolic intermediates bind non-enzymatically to intracellular proteins to produce AGEs. Among these AGEs, GA-AGEs, also known as TAGE, are highly cytotoxic and have been implicated in various pathological conditions [3], such as diabetic complications, insulin resistance, heart disease, Alzheimer’s disease, hypertension, nonalcoholic steatohepatitis, obesity, and cancer. GA-AGEs predominantly form intracellularly as a result of glycation reactions between GA and intracellular proteins, resulting in a wide range of GA-AGE molecules with different sizes and properties. The mechanisms by which GA-AGEs exert their cytotoxic effects have not yet been elucidated in detail; however, previous studies suggested that GA-AGE toxicity was attributed to oxidative stress damage, the loss of protein functions due to glycation modifications, and the aggregation and accumulation of GA-AGEs [3].

To the best of our knowledge, this is the first study to show the rapid degradation (less than 2 h) of a protein in an experimental model in which intracellular AGEs were formed by the administration of glycating agents, such as GA and methylglyoxal (MGO), to cells. Even when d270KD was fused with relatively stable proteins, such as EGFP, ZsGreen1, or luciferase, the GA-responsive rapid degradation property was retained (Figure 3, Figure 4, and Appendix A), indicating that d270KD functions as a degradation element induced by glycated substances. Furthermore, a reporter assay using luciferase activity in d270KD-Luc-transfected cells allowed the degradation process to be monitored without a Western blot analysis, and GA-induced reductions in luciferase activity were attenuated by inhibitors of AGE formation in a concentration-dependent manner (Figure 4c). Therefore, this reporter assay using d270KD-Luc-transfected cells may be a useful tool for the rapid screening of natural compounds that inhibit the formation of AGEs.

The etiology of GA-AGE toxicity is attributed to the functional impairment of the proteins themselves as a result of GA-AGE modifications. Previous studies demonstrated that chaperone molecules, such as Hsc70, and a mediator of apoptosis, caspase-3, formed high-molecular-weight complexes that led to a loss of activity due to GA-AGE modifications [3]. In the present study, we observed the similar formation and accumulation of high-molecular-weight complexes of CHK1 after the GA stimulation, as shown in Figure 1. CHK1 is a well-known key player in the cellular stress response to DNA damage and the normal progression of the cell cycle under non-stress conditions [27]. Therefore, CHK1 may be a target of GA-AGE modifications, and its functional impairment may be responsible for the induction of GA-AGE-mediated cell death. Notably, when HeLa cells expressing Flag-CHK1 were stimulated with GA, a decrease in CHK1-CP levels was observed, suggesting that N-terminal CHK1 kinase fragments were susceptible to glycation modifications by GA (Figure 1b). Consistent with this result, d270WT (a kinase fragment containing the N-terminal 270-amino acid region of wild-type CHK1), which mimics CHK1-CPs, also showed time-dependent proteolysis upon GA treatment, but more slowly than d270KD (Figure 2a). In the course of our research, d270KD demonstrated accelerated degradation in response to GA compared to d270WT. However, the precise underlying reasons for this phenomenon remain unclear at present. One plausible explanation is that the structural features of d270KD make it more susceptible to ubiquitin modification than its wild-type counterpart, potentially leading to rapid degradation through the proteasome pathway. 

As Halder et al. previously reported [28], CHK1-CPs are formed through cleavage by SPRTN metalloprotease, and in our cell model, the knockdown of SPRTN by the transfection of a specific siRNA reduced the basal expression levels of Flag-CHK1-CPs (Figure 1c). While Flag-CHK1-CPs were easily detectable in basal state cells, the expression levels of endogenous CHK1-CPs were so low that only a few of their protein bands were detectable after prolonged exposure in our Western blot analysis. The pretreatment with the proteasome inhibitor MG132 increased multiple bands of CHK1-CPs to readily detectable levels (Figure 5a), suggesting that the ubiquitin–proteasome pathway is involved in regulating the basal expression levels of CHK1-CPs. The current findings, which reveal an increase in ubiquitin modification of d270KD upon GA stimulation (Figure 6b), emphasize the importance of identifying the ubiquitin ligase involved in this modification for a comprehensive understanding of the degradation mechanism of d270KD. 

A previous study reported that Mule/HUWE1, a HECT-type ubiquitin E3 ligase, ubiquitinated CHK1 and played a critical role in regulating the basal level of CHK1 [29]. In our HeLa cell model, we noted a decrease in endogenously and transiently expressed CHK1 turnover after the elimination of Mule, as shown in Figure 7a,b. However, the knockdown of Mule did not affect the degradation of d270KD in response to the GA stimulation or suppress the formation of high-molecular-weight complexes, as shown in Figure 7c. These results suggest that the degradation mechanism of d270KD, which is dependent on glycation modifications, involves a pathway distinct from the steady-state degradation mechanism of CHK1.

An important result from our Western blot analysis is that Mule was also affected by GA-induced glycation, as shown in Figure 7c (Control siRNA). Specifically, the GA stimulation for 6 h resulted in a conspicuous smear of the putative monomer form of Mule. However, it remains unclear whether the decrease in single bands was a consequence of GA-AGE-modified Mule degradation induced by the GA stimulation or the formation of high-molecular-weight complexes. This uncertainty is attributed to the technical challenges associated with analyzing the behavior of GA-AGE-modified Mule over time because the transfer efficiency of high-molecular-weight proteins (more than 460 kDa) to a PVDF membrane is very low in a conventional Western blot analysis. The loss of function of ubiquitin ligases due to GA-AGE modifications may lead to the accumulation of proteins that need to be degraded, further exacerbating non-enzymatic glycation reactions and disrupting protein homeostasis. Therefore, a more comprehensive and detailed analysis of the effects of GA-induced glycation on ubiquitin ligases is warranted in the future.

Another possible cause of cellular damage due to GA-AGEs is proteopathy, a broad term that refers to the crosslinking of glycated proteins between molecules over time, resulting in the formation of larger high-molecular-weight complexes that gradually aggregate and accumulate, disrupting normal protein function [3]. The proper folding of intracellular proteins is crucial for their biological functions, while aberrantly folded proteins may accumulate and aggregate in cells, leading to cellular stress. In response to this stress, cells activate mechanisms to monitor protein folding, such as the unfolded protein response in the endoplasmic reticulum [30], which induces autophagy, an intracellular degradation mechanism that recycles damaged proteins and protects cells [31,32,33,34]. Non-enzymatic glycation modifications to intracellular proteins may lead to structural mutations and the formation of abnormal proteins, which may be recognized as aberrant by intracellular protein quality control mechanisms.

Extracellular AGEs have also been shown to activate intracellular autophagy pathways, which often have a cytoprotective role [30,31,32,33,34]. Intracellularly formed AGEs were previously found to induce autophagy [17]. For example, in a model in which cells were treated with MGO, a glycating substance, to accumulate intracellular AGEs (MG-H1), autophagy was induced through a pathway that involved p62. Importantly, while the loss of p62 promoted the accumulation of AGEs, p62 itself appeared to be affected by MGO-induced glycation over time, forming high-molecular-weight complexes and losing its function. In our experimental model of the formation of AGEs, in the RIPA-insoluble fraction, a slow, time-dependent increase in the expression level of a single band of p62 was observed in the Western blot analysis (Figure 8a). In contrast, in the RIPA-soluble fraction, a gradual decrease was noted in the level of p62 starting 8 h after the GA stimulation. These results suggest that p62, which was present in the RIPA-soluble fraction, bound to the protein aggregates that gradually formed due to the GA stimulation and then migrated with them into the RIPA-insoluble fraction. In other words, the autophagy–lysosome pathway (and possibly the ubiquitin–proteasome degradation pathway) remains functional in the early stages of intracellular AGE formation, allowing for the proper clearance of AGEs. However, as glycation reactions continue, degradation pathways and other potential protective factors gradually transform into AGEs, resulting in the dysfunction of the AGE clearance pathway. This late stage of AGE formation may eventually induce apoptosis or passive, unprogrammed cell death by necrosis in affected cells [3]. To the best of our knowledge, the induction of autophagy in the early stages of GA-AGE formation in cells has not yet been demonstrated.

We previously reported that the stimulation of a human pancreatic beta cell line (1.4E7 cells) with GA resulted in prominent cell death, accompanied by decreases in the autophagosome markers LC3-I and LC3-II, as well as p62 [18]. These findings were observed after a prolonged stimulation (24 h) with 2 mM GA, at which stage, cell death was already induced. Therefore, important factors involved in various vital functions were modified by GA-AGEs, aggregated, and accumulated, suggesting that the GA-AGE degradation pathway was already disrupted in the late stage. To confirm the existence of an endogenous degradation and removal mechanism for GA-AGEs, a detailed analysis of the behavior of degradation-related factors from the initial stage of a glycation stimulation is needed. 

In the present study, we investigated the behavior of p62, a known selective autophagy receptor, over 10 h, starting 2 h after stimulation with 2 mM GA. The results obtained from the RIPA-insoluble fraction revealed a slight increase in total endogenous p62 levels 10 h after the administration of GA (Figure 8a). Since autophagy is responsible for the degradation of p62 when degrading its substrates, the accumulation of p62 is often observed when autophagy is impaired [17]. However, total p62 levels remained unchanged in both the RIPA-soluble and insoluble fractions after 6 h of the stimulation with 2 mM GA; therefore, the complete failure of autophagy had not yet occurred 6 h after the GA stimulation. p62 plays a key role in substrate degradation by recruiting ubiquitinated cargo to autophagosomes [35,36]. The phosphorylation of the 403rd serine (Ser403) in the UBA domain of p62 by casein kinase-2 or TANK-binding kinase 1 increased its binding affinity to the ubiquitin chain, thereby facilitating selective autophagy [37,38]. 

Interestingly, after GA stimulation for 2 h, a gradual decrease in the phosphorylated form of p62 Ser403 was observed in the RIPA-soluble fraction, in contrast to an increase over time in the RIPA-insoluble fraction (Figure 8a). Most of the d270KD high-molecular-weight complexes induced by the GA stimulation were present in the RIPA-insoluble fraction, and the level of d270KD high-molecular-weight complexes in this fraction peaked at approximately 6 h of GA stimulation, followed by a gradual decrease (Figure 8a). Furthermore, the RIPA-insoluble fraction of the cells that did not express d270KD but were stimulated after 10 h of GA treatment and also showed an abundance of high-molecular-weight complexes that were positive for anti-TAGE antibodies (Figure 8b). This result suggests that intracellular proteins modified with TAGEs (GA-AGEs) due to GA exposure form aggregates that are resistance to detergents, such as SDS, and are more likely to become insoluble intracellularly. Furthermore, the binding of phosphorylated p62 to ubiquitinated GA-AGEs may have been increased in the insoluble fraction, suggesting that the degradation system that uses selective autophagy was still partially functional during the initial stage of the GA stimulation, despite the formation of GA-AGE-modified aggregates. Importantly, immunoprecipitation experiments on the RIPA-insoluble fraction after the GA stimulation showed that the high-molecular-weight complexes of d270KD and p62 coprecipitated, and these complexes were also positive for anti-TAGE antibodies (Figure 10). These results suggest that the high-molecular-weight complexes of d270KD in the RIPA-insoluble fraction were modified by GA-AGEs and also that p62 was one of its components.

Overall, the present results highlight the importance of analyzing the behavior of degradation-related factors, such as p62, from the initial stage of glycation stimulation to elucidate the endogenous GA-AGE degradation and removal mechanisms. Further research is needed to clarify the complex interplay between GA-AGE formation, autophagy, and cellular responses to proteopathy, which will provide a more detailed understanding of the pathogenesis of GA-AGE-related cellular damage. Since proteasome inhibitors completely suppressed the GA-stimulated rapid degradation of d270KD (Figure 6a), the ubiquitin–proteasome pathway may also work in conjunction with the autophagy pathway during the early stages of GA-AGE formation to maintain intracellular protein homeostasis, which is disrupted by glycation modifications. The knockdown of p62 did not affect the degradation of d270KD induced by GA, but modestly reduced the formation of high-molecular-weight complexes derived from d270KD in the insoluble fraction (Figure 9). Therefore, p62 may play a partial role in the formation of GA-AGE aggregates in the insoluble fraction. 

We also investigated the effects of chloroquine on autophagy. The results obtained revealed that the chloroquine treatment increased the basal level of p62 in the RIPA-insoluble fraction (Figure 12b), indicating the potential inhibition of autophagy. However, the chloroquine treatment did not affect the GA-stimulated degradation of d270KD, but slightly suppressed the formation of high-molecular-weight complexes of d270KD in the RIPA-insoluble fraction, similar to the knockdown effect of p62 (Figure 11 and Figure 12). These results suggest that chloroquine interfered with the proper formation of high-molecular-weight complexes of d270KD, which is consistent with the knockdown effect of p62. One hypothesis is that p62 may be involved in sequestering randomly formed GA-AGEs throughout the cell by aggregating them, thereby preventing interference with the function of normal proteins and potentially protecting the cell. A more detailed analysis is needed on the subcellular localization of ubiquitinated and insoluble GA-AGE aggregates. However, it is important to note that the complete inhibition of autophagy by chloroquine was not fully confirmed in the present study. Therefore, the exact biological significance of the involvement of p62 in the formation of high-molecular-weight complexes of d270KD induced by GA stimulation remains to be elucidated. Further investigations are warranted to obtain a more detailed understanding of the autophagy-related effects of chloroquine in our experimental context.

## 5. Conclusions

Our study provides new insights into the intracellular degradation mechanism of GA-AGEs, offering potential therapeutic strategies to prevent or alleviate GA-AGE-related diseases. The suggestion of a specific ubiquitin ligase that recognizes the structural changes induced by GA glycation modification adds complexity to our understanding. Future investigations should not only focus on identifying this unidentified ligase but also on unraveling its differential binding affinity for d270WT and d270KD. Further exploration in this field promises to advance the development of effective interventions for diseases associated with protein aggregation and the accumulation of AGEs, spanning conditions such as diabetes, neurodegenerative diseases, and cardiovascular disorders.

## Figures and Tables

**Figure 1 cells-12-02838-f001:**
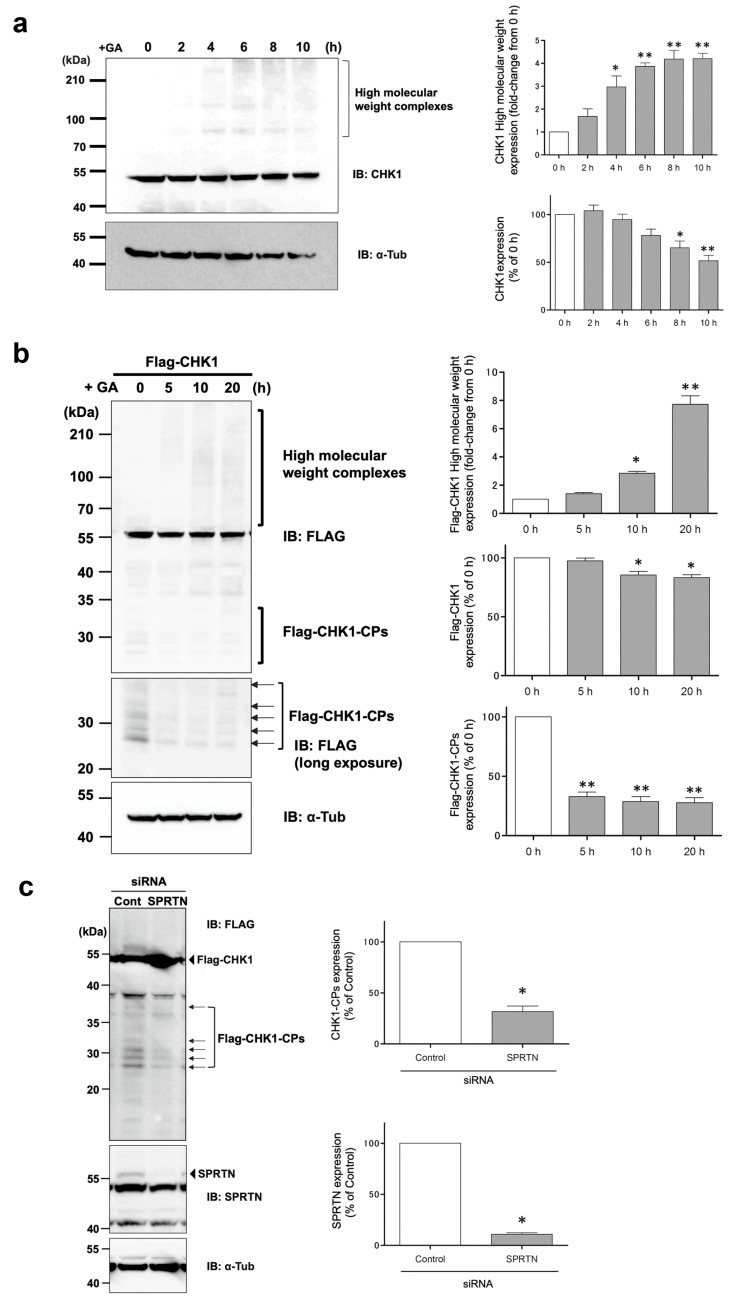
Effects of the GA treatment on GA-AGE formation with wild-type CHK1 and its N-terminal cleavage products. (**a**) HeLa cells were treated with 2 mM GA and cell lysates were collected at the indicated times for a Western blot analysis. The densitometric quantification of endogenous CHK1 high-molecular-weight complex and monomer values relative to the non-treatment group (0 h) is shown. The data were analyzed using a one-way ANOVA followed by Dunnett’s post hoc test and are shown as the mean ± SEM of three independent experiments. * *p* < 0.01, ** *p* < 0.001. (**b**) The expression vector of wild-type CHK1 (Flag-CHK1) was transfected into HeLa cells and stimulated with 2 mM GA 24 h later. Cell lysates were collected at the indicated times and subjected to a Western blot analysis. The densitometric quantification of Flag-CHK1 high-molecular-weight complex, monomer, and CP values relative to the non-treatment group is shown. The data were analyzed using a one-way ANOVA followed by Dunnett’s post hoc test and are shown as the mean ± SEM of three independent experiments. * *p* < 0.01, ** *p* < 0.001. (**c**) HeLa cells were transfected with scrambled control siRNA or SPRTN siRNA and transfected 24 h later with a Flag-CHK1 expression vector. Cell lysates were collected after 48 h and subjected to a Western blot analysis using the indicated antibodies. The densitometric quantification of Flag-CHK1 values relative to the control group (Control) is shown. The data were analyzed using the paired Student’s *t*-test and are shown as the mean ± SEM of three independent experiments. * *p* < 0.001. The positions of the bands corresponding to CHK1-CPs are indicated by arrows.

**Figure 2 cells-12-02838-f002:**
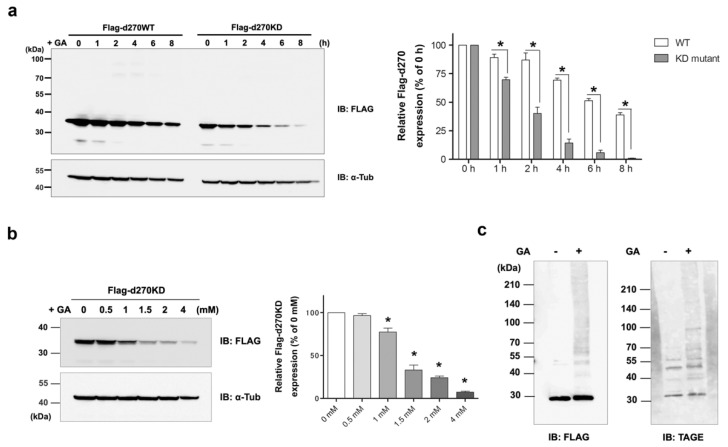
GA-stimulated rapid protein degradation of intracellular C-terminal deletion mutants of CHK1. (**a**) A CHK1 mutant expression vector lacking the C-terminal regulatory domain (Flag-d270WT) or its constitutive kinase-dead mutant expression vector (Flag-d270KD) was transfected into HeLa cells. Cell lysates were collected at the indicated times after 2 mM GA was applied to cells 24 h later. Lysates were subjected to a Western blot analysis using the indicated antibodies. The densitometric quantification of Flag-d270KD values relative to the non-treatment group (0 h) is shown. The data were analyzed using a two-way ANOVA followed by the Bonferroni post hoc test and are shown as the mean ± SEM of three independent experiments. * *p* < 0.001. (**b**) A Flag-d270KD expression vector was transfected into HeLa cells and cells were stimulated 24 h later with the indicated concentration of GA for 6 h. A Western blot analysis was performed on lysates collected from these cells as described above. The densitometric quantification of Flag-d270KD values relative to the non-treatment group (0 mM) is shown. The data were analyzed using a one-way ANOVA followed by Dunnett’s post hoc test and are shown as the mean ± SEM of three independent experiments. * *p* < 0.001. (**c**) In vitro GA-AGE modifications to the d270KD recombinant protein. A Flag-d270KD expression vector was transfected into HeLa cells and cell lysates were extracted 24 h later. The lysates were immunoprecipitated with an anti-Flag antibody to purify the Flag-d270KD recombinant protein, which was then reacted with 4 mM GA in vitro for 20 h. GA-AGE modifications to the d270KD recombinant protein were analyzed by Western blotting.

**Figure 3 cells-12-02838-f003:**
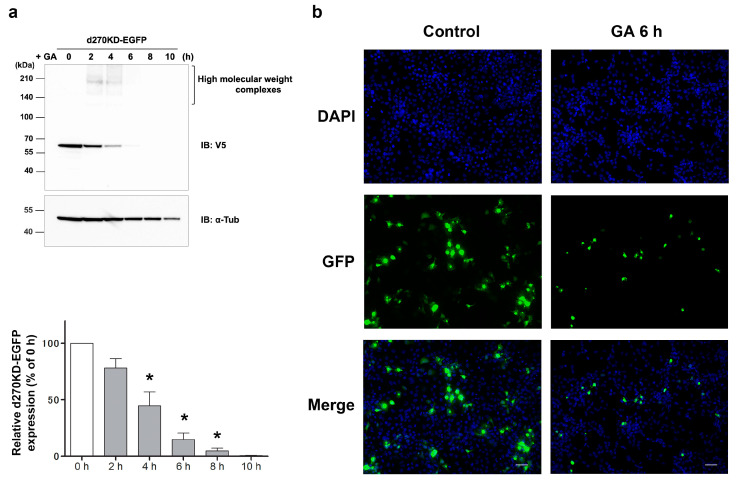
GA-stimulated rapid intracellular degradation of d270KD-EGFP fusion protein. (**a**) The expression vector of d270KD fused to EGFP (d270KD-EGFP) was transfected into COS-7 cells and stimulated with 4 mM GA 48 h later. Cell lysates were collected at the indicated times and subjected to a Western blot analysis. The densitometric quantification of d270KD-EGFP values relative to the non-treatment group (0 h) is shown. The data were analyzed using a one-way ANOVA followed by Dunnett’s post hoc test and are shown as the mean ± SEM of three independent experiments. * *p* < 0.001. (**b**) COS-7 cells were transfected with the d270KD-EGFP expression vector for 24 h, stimulated with phosphate buffer (control) or 4 mM GA for 6 h, and then fixed. Nuclei were counterstained with DAPI and analyzed by fluorescence microscopy. Scale bars = 50 µm.

**Figure 4 cells-12-02838-f004:**
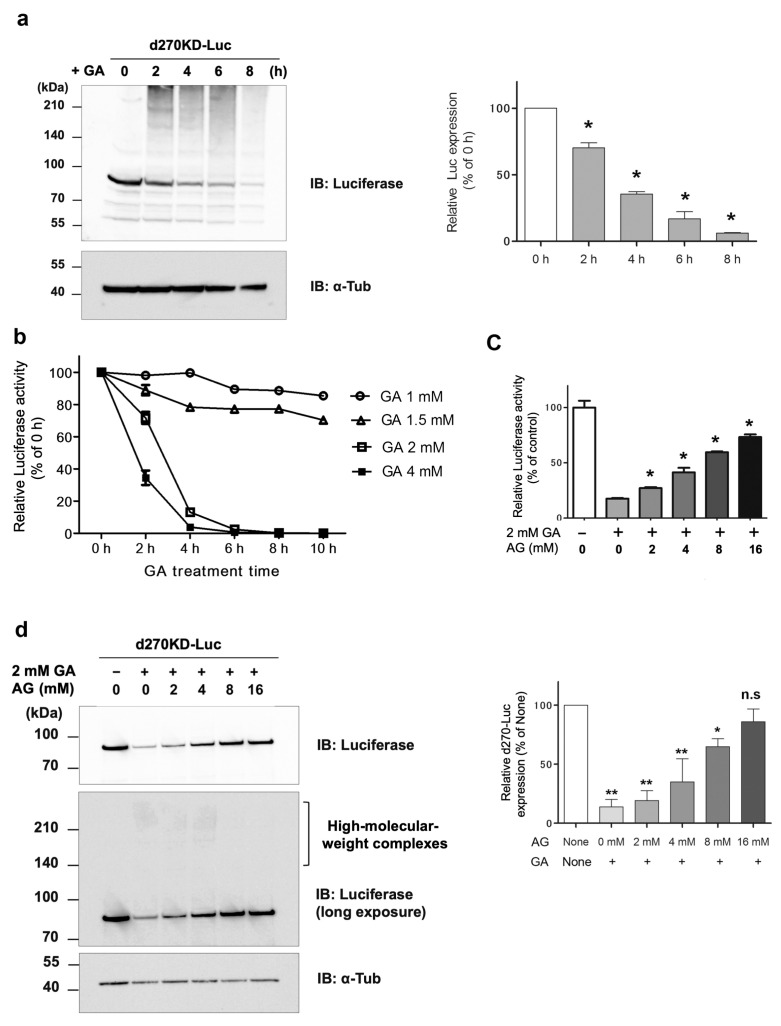
Rapid proteolysis of d270KD-fused luciferase and its reduced activity upon GA stimulation. (**a**) A d270KD fused to luciferase expression vector (d270KD-Luc) was transfected into HeLa cells and stimulated with 4 mM GA after 24 h. Cell lysates were collected at the indicated times and subjected to a Western blot analysis. The densitometric quantification of d270KD-Luc values relative to the non-treatment group (0 h) is shown. The data were analyzed using a one-way ANOVA followed by Dunnett’s post hoc test and are shown as the mean ± SEM of three independent experiments. * *p* < 0.001. (**b**) As in (**a**), d270KD-Luc was transfected into HeLa cells for 24 h and stimulated with GA at the indicated concentrations. Cell lysates collected at each indicated time were subjected to the luciferase assay. (**c**) HeLa cells were transfected with d270KD-Luc for 24 h and then pretreated with phosphate buffer or aminoguanidine (AG) at the indicated concentrations for 2 h. The cells were then stimulated with phosphate buffer or 2 mM GA for 6 h and lysed. Cell lysates were collected and luciferase activity was measured and expressed as 100 for the control. Each value was obtained from three independent experiments. Error bars indicate SEM. * *p* <0.01 versus 2 mM GA alone. (**d**) Cell lysates treated similarly to the experiment in (**c**) were subjected to a Western blot analysis using the antibodies indicated. The densitometric quantification of d270KD-Luc values relative to the non-treatment group (None) are shown. The data were analyzed using a one-way ANOVA followed by Dunnett’s post hoc test and are shown as the mean ± SEM of three independent experiments. * *p* < 0.01, ** *p* < 0.001, ns means not significant.

**Figure 5 cells-12-02838-f005:**
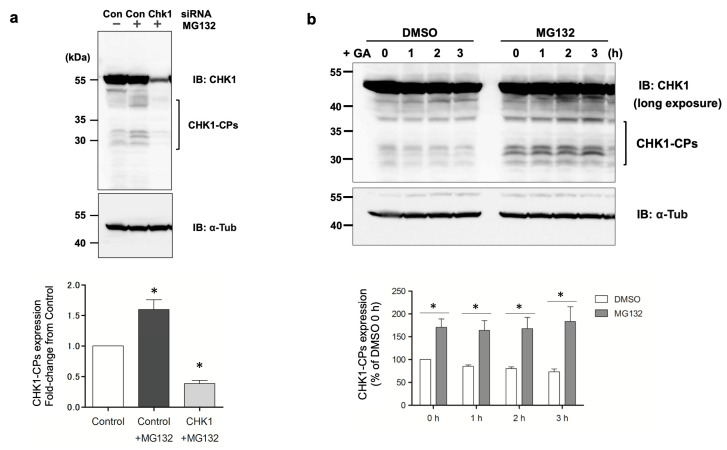
Involvement of the proteasome pathway in the regulation of CHK1-CPs at the basal level. (**a**) HeLa cells were transfected with scrambled control siRNA (Con) or Chk1 siRNA, treated with 5 mM of the proteasome inhibitor MG132 (+) or DMSO (−) 48 h later, and cell lysates were collected after 5 h. The resulting cell lysates were subjected to a Western blot analysis with the indicated antibodies. The densitometric quantification of endogenous CHK1-CP values relative to the non-treated control group (Con) is shown. The data were analyzed using a one-way ANOVA by Dunnett’s post hoc test and are shown as the mean ± SEM of three independent experiments. * *p* < 0.01. (**b**) HeLa cells were treated with 5 mM of the proteasome inhibitor MG132 or DMSO for 4 h. Cells were treated with 2 mM GA and cell lysates were collected at the indicated times for a Western blot analysis. The densitometric quantification of endogenous CHK1-CP values relative to the DMSO-treated control group (DMSO 0 h) is shown. The data were analyzed using a two-way ANOVA followed by the Bonferroni post hoc test and are shown as the mean ± SEM of three independent experiments. * *p* < 0.001.

**Figure 6 cells-12-02838-f006:**
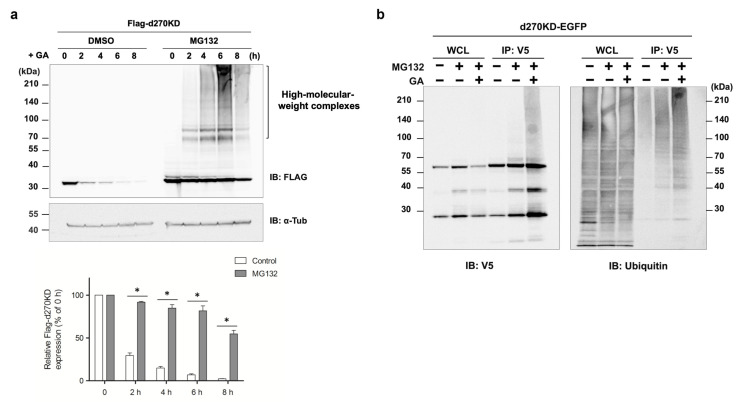
Involvement of the proteasome pathway in the rapid degradation of d270KD by GA. (**a**) HeLa cells were transfected with the Flag-d270KD expression vector and cells were incubated 24 h later with 5 mM of a proteasome inhibitor (MG132) or DMSO (control) for 4 h. Cells were treated with 4 mM GA and cell lysates were collected at the indicated times for a Western blot analysis. The densitometric quantification of Flag-d270KD values relative to the non-treatment group (0 h) is shown. The data were analyzed using a two-way ANOVA followed by the Bonferroni post hoc test and are shown as the mean ± SEM of three independent experiments. * *p* < 0.001. (**b**) The GA stimulation increased the ubiquitination of d270KD-fused EGFP. The expression vector for the d270KD fused to EGFP (d270KD-EGFP) expression vector was transfected into HeLa cells for 48 h and the cells were then treated with DMSO (control) or 5 µM MG132 for 3 h. Cells were collected after the stimulation with phosphate buffer or 4 mM GA for 3 h. Cell lysates were extracted under denaturing conditions, diluted in lysis buffer, and immunoprecipitated (IP) with an anti-V5 antibody. Whole cell lysates (WCL) and IP samples were subjected to a Western blot analysis using the indicated antibodies.

**Figure 7 cells-12-02838-f007:**
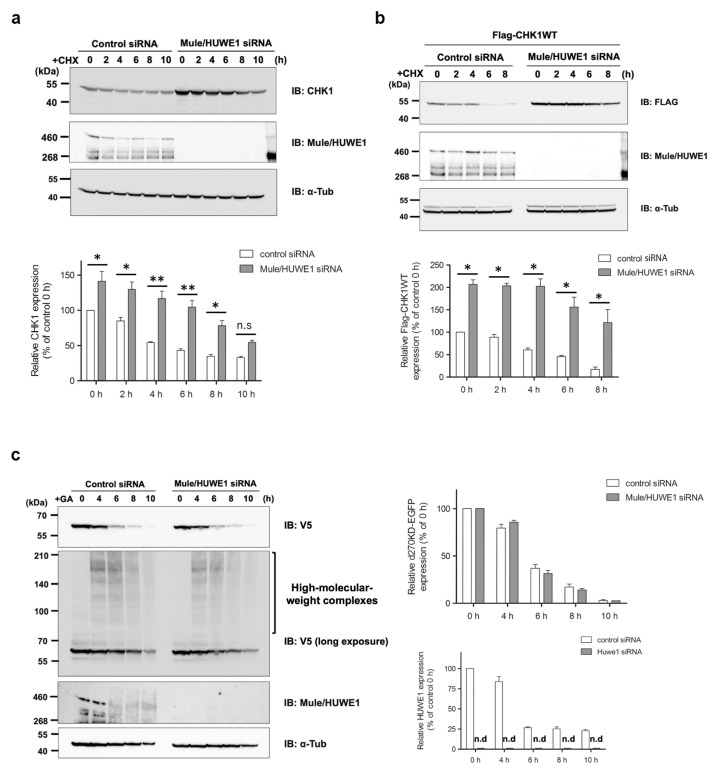
Effects of the elimination of Mule/HUWE1 E3 ubiquitin ligase on the rapid degradation of d270KD stimulated by GA (**a**) HeLa cells were transfected with scrambled control siRNA (control) or Mule/HUWE1 siRNA, treated with cycloheximide (CHX; 20 µg/mL) 48 h later, and cell lysates were collected at the indicated times. The resulting cell lysates were subjected to a Western blot analysis with the indicated antibodies. (**b**,**c**) Scrambled control siRNA or Mule/HUWE1 siRNA was transfected into HeLa cells as in (**a**), and cells were transfected 24 h later with a Flag-CHK1WT (**b**) or d270KD-EGFP expression vector (**c**). Cells were treated 24 h later with cycloheximide (CHX; 20 µg/mL) (**b**) or 2 mM GA (**c**) for the indicated times, and cell lysates were collected for a Western blot analysis. The densitometric quantification of endogenous CHK1 (**a**), Flag-CHK1WT (**b**), or d270KD-EGFP (**c**) values relative to the non-treatment group (0 h) is shown. The data were analyzed using a two-way ANOVA followed by the Bonferroni post hoc test and are shown as the mean values ± SEM of three independent experiments. * *p* < 0.01, ** *p* < 0.001, n.s and n.d mean not significant and not detected, respectively.

**Figure 8 cells-12-02838-f008:**
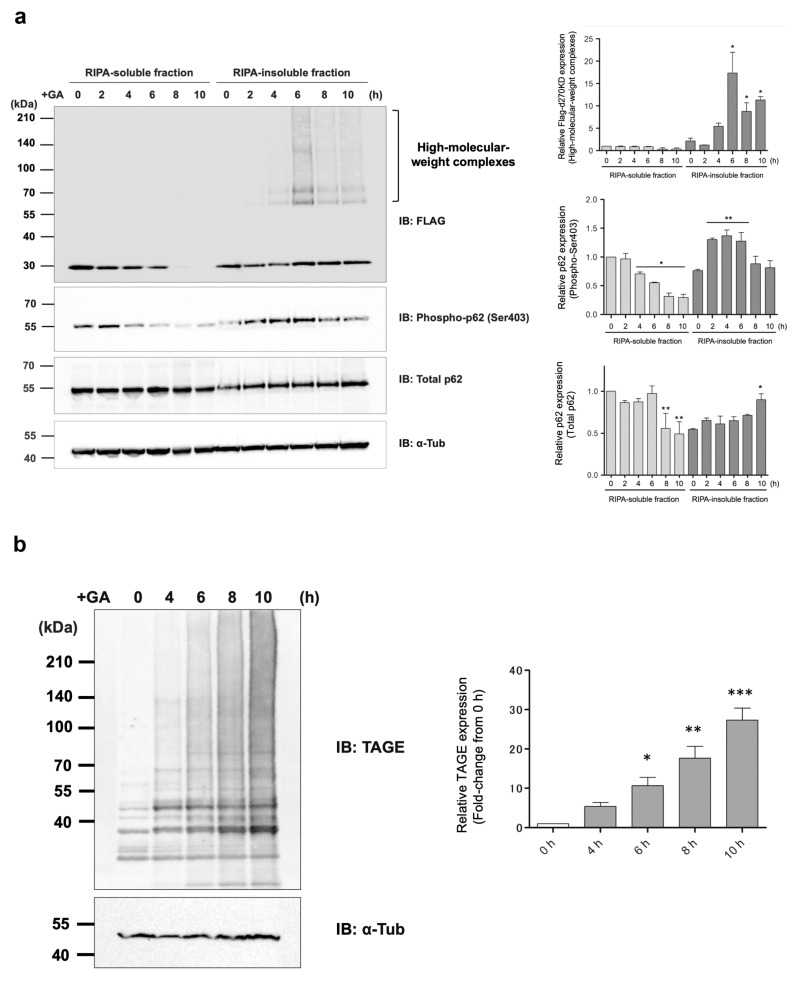
All high-molecular-weight complexes of d270KD formed by a GA stimulation accumulate in the insoluble fraction of RIPA. (**a**) A Flag-d270KD expression vector was transfected into HeLa cells, and cells were stimulated 24 h later with 2 mM GA and collected at the indicated times. Cells were lysed in RIPA buffer and separated into supernatant (RIPA-soluble fraction) and pellet (RIPA-insoluble fraction) by centrifugation. After the addition of SDS sample buffer to each fraction, samples from both fractions were analyzed by Western blotting using the antibodies indicated. The densitometric quantification of Flag-d270KD high-molecular-weight complex, p62 (phosphor-Ser403), and p62 (total) values relative to the non-treatment RIPA-soluble fraction group are shown. The data were analyzed using a two-way ANOVA followed by the Bonferroni post hoc test and are shown as the mean ± SEM of three independent experiments. * *p* < 0.05, ** *p* < 0.01 vs. 0 h. (**b**) HeLa cells were treated with phosphate buffer (control) or 2 mM GA. Cells were collected at the indicated times and lysed in RIPA buffer. Each cell lysate was separated into RIPA-soluble/-insoluble fractions in the same manner as in (**a**). Samples of RIPA-insoluble fractions were prepared by three independent experiments and subjected to a Western blot analysis using the indicated antibodies. The densitometric quantification of TAGE values relative to the non-treatment fraction group (0 h) is shown. The data were analyzed using a one-way ANOVA followed by Dunnett’s post hoc test and are shown as the mean ± SEM of three independent experiments. * *p* < 0.05, ** *p* < 0.01, *** *p* < 0.001.

**Figure 9 cells-12-02838-f009:**
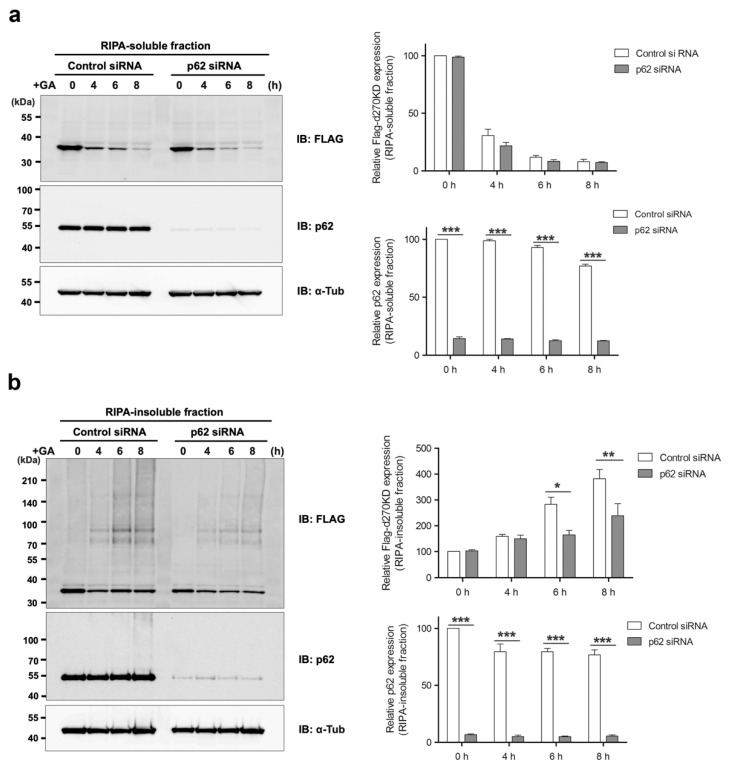
Effects of the p62/SQSTM1 knockdown on the rapid GA-stimulated degradation of d270KD. HeLa cells were transfected with scrambled control siRNA or p62/SQSTM1 siRNA and transfected 24 h later with a Flag-d270KD expression vector. Cells were stimulated 24 h later with 2 mM GA for the indicated times. RIPA buffer was added to cells and cell lysates were separated into RIPA-soluble (**a**) and RIPA-insoluble (**b**) fractions by centrifugation. Each fraction was analyzed by Western blotting using the indicated antibodies. The densitometric quantification of Flag-d270KD and p62 (total) values relative to the non-treatment group (0 h) is shown. The data were analyzed using a two-way ANOVA followed by the Bonferroni post hoc test and are shown as the mean ± SEM of three independent experiments. * *p* < 0.05, ** *p* < 0.01, *** *p* < 0.001.

**Figure 10 cells-12-02838-f010:**
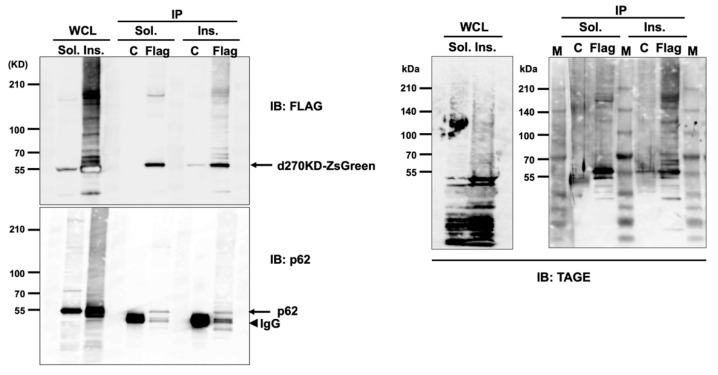
p62 is a component of GA-AGE-modified d270KD high-molecular-weight complexes present in the RIPA-insoluble fraction after GA stimulation. The expression vector for the Flag-d270KD fused to ZsGreen1 (d270KD-ZsGreen) expression vector was transfected into HeLa cells for 48 h and cells were then treated with 2 mM GA for 3 h. RIPA buffer was added to the cells and the cell lysates were separated into RIPA-soluble (Sol.) and RIPA-insoluble (Ins.) fractions by centrifugation. Each fraction was immunoprecipitated (IP) with an anti-FLAG antibody (Flag) or normal rabbit IgG (C). Whole cell lysates (WCL) and IP samples were subjected to a Western blot analysis using the indicated antibodies. M, protein size marker.

**Figure 11 cells-12-02838-f011:**
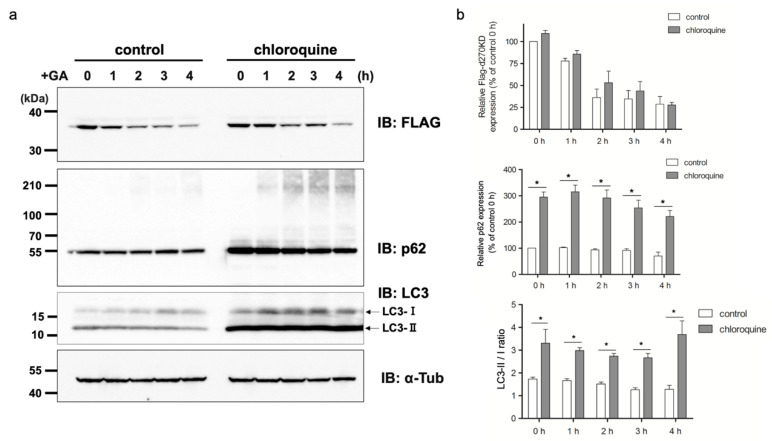
Effects of chloroquine, an autophagy inhibitor, on the rapid GA-stimulated degradation of d270KD. (**a**) HeLa cells were transfected with a Flag-d270KD expression vector. The cells were stimulated 24 h later with 50 µM chloroquine for 20 h. The cells were collected after the stimulation with phosphate buffer or 2 mM GA for the indicated times. The resulting cell lysates were subjected to a Western blot analysis with the indicated antibodies. (**b**) The densitometric quantification of Flag-d270KD and total p62 values relative to the non-treatment control group (control 0 h) are shown. The data were analyzed using a two-way ANOVA followed by the Bonferroni post hoc test and are shown as the mean ± SEM of three independent experiments. * *p* < 0.001.

**Figure 12 cells-12-02838-f012:**
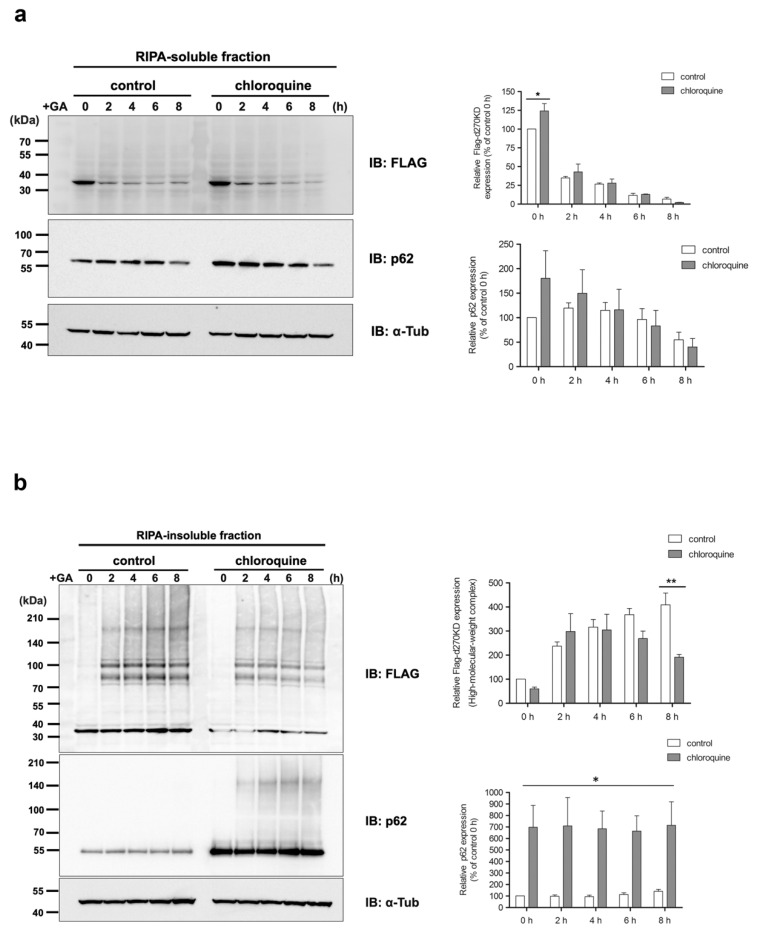
Effects of chloroquine on the rapid GA-stimulated degradation of d270KD in the RIPA-soluble and RIPA-insoluble fractions. HeLa cells were transfected with a Flag-d270KD expression vector. The cells were stimulated 24 h later with 50 µM chloroquine for 20 h. The cells were collected after the stimulation with phosphate buffer or 2 mM GA for the indicated times. RIPA buffer was added to cells and cell lysates were separated into RIPA-soluble (**a**) and RIPA-insoluble (**b**) fractions by centrifugation. Each fraction was analyzed by Western blotting using the indicated antibodies. The densitometric quantification of Flag-d270KD and total p62 values relative to the non-treatment control group (control 0 h) is shown. The data were analyzed using a two-way ANOVA followed by the Bonferroni post hoc test and are shown as the mean ± SEM of three independent experiments. * *p* < 0.001 (**a**); * *p* < 0.05, ** *p* < 0.01 (**b**).

## Data Availability

Data for this study can be found in the main article or in the Appendix A.

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
