# Peer review of "A Novel Approach: Investigating the Intracellular Clearance Mechanism of Glyceraldehyde-Derived Advanced Glycation End-Products Using the Artificial Checkpoint Kinase 1 d270KD Mutant as a Substrate Model"

_cells, 2023, doi:10.3390/cells12242838_

Round 1
Reviewer 1 Report (Previous Reviewer 2)
Comments and Suggestions for Authors
This paper has become excellent after two rounds of revision. It is worthy of publication.
Author Response
Please see the attachment.

Reviewer 2 Report (New Reviewer)
Comments and Suggestions for Authors
The intracellular behavior of Checkpoint kinase 1 cleavage product d270KD stimulated by glyceraldehyde was studied by Western blot and immunoprecipitation. It was found that d270KD can be rapidly degraded by ubiquitin-proteasome pathway and co-precipitate with selective autophagy cargo acceptor p62 to form high molecular weight insoluble complex. This suggests that the ubiquitin-proteasome pathway and autophagy may synergically participate in the clearance of Glycation products of glyceraldehyde (GA-AGEs) in the cell. This study laid a foundation for exploring the intracellular degradation mechanism of GA-AGEs. There is room to optimize the framework and logic of this manuscript as well as the diagrams. This manuscript could be accepted after minor revision. Other questions were shown below:
1. At the beginning, we can add more background knowledge about the common sources, types and hazards of AGEs in food, so that readers can have a fuller understanding of the research object. Please refer this refence (Food chemistry, 385(2022): 132697. Food Chemistry, 417(2023):135861.).
2. The structure, formation mechanism and biological effects of GA-AGEs can be described more specifically and in detail. Please refer this reference (Critical Reviews in Food Science and Nutrition, 2023, 63(29), 9816–9842.).
3. Failure to state the purity and source of the GA and AG you used may affect the reproducibility and comparability of your experimental results.
4. More methods can be used to identify and quantify your GA-AGEs, such as high performance liquid chromatography-tandem mass spectrometry (HPLC-MS/MS) or fluorescence spectrometry.
5. Page 7, the arrows in Figure 1 should indicate the size of the corresponding CHK1 cleavage product for easy identification by the reader.
6. The fluorescence microscope image in Figure 3 should have been taken with the same magnification and exposure time.
7. The paper did not put forward specific suggestions and prospects for the future research direction, such as how to further reveal the mechanism of the influence of GA-AGEs on CHK1 function.
8. In the discussion section, it is not explained in detail why d270WT and d270KD have different degradation and modification modes under GA stimulation.
The reference should be updated in recent years.
Comments on the Quality of English LanguageThe intracellular behavior of Checkpoint kinase 1 cleavage product d270KD stimulated by glyceraldehyde was studied by Western blot and immunoprecipitation. It was found that d270KD can be rapidly degraded by ubiquitin-proteasome pathway and co-precipitate with selective autophagy cargo acceptor p62 to form high molecular weight insoluble complex. This suggests that the ubiquitin-proteasome pathway and autophagy may synergically participate in the clearance of Glycation products of glyceraldehyde (GA-AGEs) in the cell. This study laid a foundation for exploring the intracellular degradation mechanism of GA-AGEs. There is room to optimize the framework and logic of this manuscript as well as the diagrams. This manuscript could be accepted after minor revision. Other questions were shown below:
1. At the beginning, we can add more background knowledge about the common sources, types and hazards of AGEs in food, so that readers can have a fuller understanding of the research object. Please refer this refence (Food chemistry, 385(2022): 132697. Food Chemistry, 417(2023):135861.).
2. The structure, formation mechanism and biological effects of GA-AGEs can be described more specifically and in detail. Please refer this reference (Critical Reviews in Food Science and Nutrition, 2023, 63(29), 9816–9842.).
3. Failure to state the purity and source of the GA and AG you used may affect the reproducibility and comparability of your experimental results.
4. More methods can be used to identify and quantify your GA-AGEs, such as high performance liquid chromatography-tandem mass spectrometry (HPLC-MS/MS) or fluorescence spectrometry.
5. Page 7, the arrows in Figure 1 should indicate the size of the corresponding CHK1 cleavage product for easy identification by the reader.
6. The fluorescence microscope image in Figure 3 should have been taken with the same magnification and exposure time.
7. The paper did not put forward specific suggestions and prospects for the future research direction, such as how to further reveal the mechanism of the influence of GA-AGEs on CHK1 function.
8. In the discussion section, it is not explained in detail why d270WT and d270KD have different degradation and modification modes under GA stimulation.
The reference should be updated in recent years.
Round 2
Reviewer 2 Report (New Reviewer)
Comments and Suggestions for Authors
The manuscript has been responded to according to the reviewer's comment. It can be accepted in the current revision.
Comments on the Quality of English LanguageThe manuscript has been responded to according to the reviewer's comment. It can be accepted in the current revision.
This manuscript is a resubmission of an earlier submission. The following is a list of the peer review reports and author responses from that submission.
Round 1
Reviewer 1 Report
Comments and Suggestions for Authors
In the manuscript “The Ubiquitin-Proteasome Pathway and p62/SQSTM1 are Involved in The Rapid Degradation and Aggregation Mechanisms of CHK1 Mutant-Derived Toxic AGEs Formed by Glyceraldehyde.”, Kenji Takeda et al. tried to explore degradative mechanism(s) involved in the clearance of CHK1 mutant d270KD. Using mostly westerblotting the authors claim that the degradation is “mainly” mediated by ubiquitin-proteasome system (although lysosomal degradation was not evaluated). Although the topics in this study could be of interest for the journal’s readers, the research design is limited and the conclusions drawn by the authors are overinterpretation of the data (see below). Although the results are clearly presented, there is a lack of quantification in most of the experiments. The manuscript will require major revision and some critical experiments should be carried out to confirm if the conclusion are right (or not).
1. The authors used the term “Toxic advanced glycation end-products (TAGE)”. It is not a common term in the glycation field (only 22 results were found in Pubmed and most of the experimental papers were generated by the authors of the submitted manuscript; see https://pubmed.ncbi.nlm.nih.gov/?term=%22toxic+advanced+glycation+end-products%22+TAGE&sort=date ). This reviewer would suggest to change the term “Toxic advanced glycation end-products (TAGE)” to “glyceraldehyde-derived AGEs”. Although the authors mentioned "toxicity" in the manuscript, please note that there is no experiment to evaluate toxicity.
2. What is the significance of the CHK1 mutant analyzed in the study?? Has d270KD been identified in any disease? Is the d270KD mutant found in any pathological condition?? The authors did not explain why they studied specifically this mutant. Why is it biologically relevant?
3. What exactly recognizes the antibody against TAGE (previously reported in ref34?? This reviewer have read the paper where non-CML AGE antibodies were developed but the information about the antibody used in this manuscript is unclear. A detailed explanation about the antibody and which AGEs are recognized is information needed to understand the conclusions of the manuscript.
4. Proper quantification and proper stats are needed to draw any conclusion. At least, quantification +stats should be provide for figures 1A, 2A, 2B, 3A, 5A, 6A, 6B, 6C, 7A, 7B, 8A, 8B.
5. The authors mentioned “degradation” from the beginning of the text. However, the first experiment about degradation (specifically UPS-mediated degradation) appears in Fig5A. The authors did not analyze degradation in the previous figures, only total levels of the proteins in the lysates.
6. What is the rationale to use AG?? How does the inhibitor of AGEs formation impact the AGEs degradation? If the authors want to explore the impact of AG in AGES degradation should co-incubate the cells with both proteasomal and lysosomal inhibitors along with GA.
7. The experiment shown in Fig6B is incomplete. Cells treated (or not) with GA should be exposed to MG132 to evaluate the UPS-mediated degradation. Quantification and stats are missing. In addition, the same experiment should be carried out in presence of lysosomal inhibitors to evaluate potential degradation through autophagy. As mentioned by the author, both proteasome and autophagy degrade AGEs.
8. the authors claimed that “The autophagy pathway is known to be inhibited in cells stimulated with GA, along with an increase in intracellular TAGE levels.”. No evidences are show to draw this conclusion. Some evidences should be shown in the cellular models used in the study to prove that. In addition, in order to discard the lysosomal degradation of Flag-d270KD, a time-dependent experiment should be carried out in cells expressing the mutant in presence of lysosomal inhibitor, that is, the same experiment that in Fig5a but using a lysosomal inhibitor instead of MG132. In addition, it is experiment should much more informative if it is carried out in control and p62knockdown backgrounds.
9. The whole blot should be shown for antibodies against p62. The authors claim that Mule was affectd by glycation. It has been previously published that glycation also induced the accumulation of high-molecular weight p62.
Reviewer 2 Report
Comments and Suggestions for Authors
In this manuscript, the authors present that a truncated form of CHK1 (d270KD) undergoes rapid degradation and p62-dependent aggregation by GA-induced glycation. Overall, the data is insufficient to substantiate the authors' claims. Additionally, there is very little evidence provided regarding CHK1, and it is unknown whether CHK1 WT reflect data on d270KD.
In Fig1a, expression level of Flag-d270KD is extremely low. The authors need to compare modification and degradation of Flag-d270KD under similar conditions to those of other proteins. Are ubiquitinated products involved in high molecular complexes? The authors should confirm it using anti-poly-Ub antibody.
In Figs 2-4, statistical analysis is necessary. Protein turnover rates should be analyzed using graph analysis in Fig 6.
In Fig 7a, the authors demonstrate the interaction of p62 with d270KD. But, this is not shown to interact both proteins directly. This should be performed by IP.
In Fig 8b, 30kDa bands of Flag-d270KD disappeared. Why?
Reviewer 3 Report
Comments and Suggestions for Authors
Kenji Takeda and colleagues present a quality and well-written experimental manuscript focused on the ubiquitin-proteasome pathway and p62/SQSTM1 are involved in the rapid degradation and aggregation mechanisms of CHK1 mutant-derived toxic AGEs formed by glyceraldehyde.
Authors identified the checkpoint kinase-1 (CHK1) mutant, d270KD, which was rapidly degraded intracellularly by GA, and showed that its degradation was mainly mediated by the ubiquitin-proteasome pathway. The high-molecular-weight complexes formed by the GA stimulation of d270KD were abundant in the RIPA-insoluble fraction, which also contained high levels of TAGE. The knockdown of p62/SQSTM1 reduced the amount of high-molecular-weight complexes in the RIPA-insoluble fraction, indicating its involvement in the formation of TAGE aggregates.
Authors investigated the behavior of p62, a known selective autophagy receptor, over 10 h, starting 2 h after a stimulation with 2 mM GA. The results obtained revealed a slight change in total endogenous p62 levels.
Authors suggest that the ubiquitin-proteasome pathway and p62 play a role in the degradation and aggregation of intracellular TAGE formed by GA. This study provides new insights into the mechanisms underlying TAGE metabolism and may lead to the development of novel therapeutic strategies for diseases associated with TAGE accumulation.
Finally, authors conclude that they provided novel insights into the intracellular degradation mechanism of TAGE and proposes potential therapeutic strategies to prevent or mitigate TAGE-related diseases. Further research in this area will contribute to the development of effective interventions for diseases associated with protein aggregation and the accumulation of AGEs, including diabetes, neurodegenerative diseases, and cardiovascular diseases.
Overall, the manuscript is highly valuable for the scientific community and should be accepted for publication after minor edits are made.
=======================================
Other comments:
1) Please check for typos and punctuation throughout the manuscript.
2) With regards to E3 ubiquitin ligases - authors are kindly encouraged to cite the following article that reports the structural aspect of a certain E3 ubiquitin ligase family that is relevant for development of therapeutics targeting this class of proteins. DOI: 10.1371/journal.pone.0131218
Round 2
Reviewer 1 Report
Comments and Suggestions for Authors
In the new version of the manuscript, Kenji Takeda et al. improved the study including new experiments. The research design is appropriate (although this reviewer encourages the authors to explore the impact of GA in Chk1-WT (see below), the methods are well-described, results clearly presented, and conclusions supported by the results. Despite the previous paragraph, this reviewer found significance of the content quite low. Why?? Basically, because the whole study is focused on a mutant whose “existence of these mutants in nature has not been confirmed”, that is, d270KD has not been identified in any disease or pathological condition. In sum, there is a worrying lack of significance in the study if the mutant is presented as an engineered form of CHK1 that does not exist in the nature. I cannot see major technical issues and I would like to recommend the paper for publication, but the significance of the study is a major concern. I would recommend the journal to send the paper to a new reviewer (before making any decision) to get feedback about the significance of the findings.
This reviewer suggests the authors to emphasize some aspects of the study with the idea to bypass the problem.
1. The authors did not pay attention to some results that could be of interest. In Fig2 it is shown that GA stimulates rapid degradation of Flag-d270WT. At least, the authors can claim that the N-terminal region of Chk1 (amino acids 1-270) in pFlag-Chk1Wt is sensitive to GA providing some clues about the biology of an important protein (Chk1). In addition, the lack of regulatory domain (missing in the mutant) would accelerate the degradation.
2. Autophagy is not required for the degradation of the mutant d270KD (Fig.10) suggesting that, when p62 is inactive or absent, the mutant d270KD is targeted to the proteasome for degradation. In other words, the degradation of mutant d270KD is autophagy-independent.
The reviewer strongly encourages the authors to show the endogenous Chk1 or engineered whole-length Chk1 WT is sensitive to GA and the degradation takes place through proteasomal activity (but not autophagy). This is the only experiment requested. Note that it is already shown in Fig6 that CHK1 endogenous level decreased significantly upon GA treatment. If the authors show that GA-induced degradation is through proteasome (but not autophagy) would prove a strong piece of biological significance in a protein that is key in multiple cellular processes (DNA replication, mitosis, etc) and altered in pathologies such as cancer. It could somehow justify later the in-depth study in the N-terminal region of Chk1.
Recommendations:
1. The title “Roles of Ubiquitin-Proteasome and Autophagy Pathways in Intracellular Clearance of Toxic Advanced Glycation End-products Derived from Checkpoint kinase-1 Mutants” is not accurate at all. This reviewer recommends to modify the title to be more specific with the topics covered in the study. The study is exclusively focused on the degradation of the mutant d270KD
2. Some information should be included in Intro and Discussion to emphasize the significance of Glyceraldehyde-derived AGEs (GA-AGEs). For example,
“Glyceraldehyde-derived AGEs (GA-AGEs) represent a structurally heterogeneous group of molecules; therefore, the specific structures within this group that are responsible for the observed toxicity remain unclear”
“an anti-GA-AGEs antibody completely mitigated neurotoxicity induced by serum AGEs in patients with diabetic nephropathy with hemodialysis (DN-HD), while antibodies targeting other types of AGEs or CML did not exert the same protective effects” ‘only AGE structures containing epitopes recognized by the anti-GA-AGEs antibody are toxic” (DOI: 10.1093/jnen/59.12.1094)
“AGEs recognized by the anti-GA-AGE antibody as toxic-AGEs (TAGE), distinguishing them from all other AGEs and known GA-derived AGEs, such as GLAP, triosidines, and MG-H1”
3. In Methods should be included that TAGE antibody “recognized distinct epitopes from GA-derived structures, such as 3-hydroxy-5-hydroxymethyl-pyridinium (GLAP) and triosidines, but not well-characterized AGEs containing CML and Nε-(carboxyethyl)lysine (CEL) or other AGEs derived from reducing sugars and carbonyl molecules, including pyrraline, pentosidine, crossline, argpyrimidine, GO- or MGO-lysine dimers, and GO- or MGO-derived hydroimidazolone”
3. Although the authors mentioned that 2 reviews have recently mentioned the term TAGE, this term was coined over 2 decades ago and it is not a common term in the glycation field. Again, this reviewer strongly encourages the authors to use “glyceraldehyde-derived AGEs”, instead of “Toxic advanced glycation end-products (TAGE)”. The only experiment about toxicity is included as Supplementary Figure and it does not include the mutant Chk1. In other words, toxicity is not a major topic in this study, and it should not be emphasized.
4. Only an increase in p62 expression levels does not mean that autophagy is inhibited (see PMID: 33634751 ) . The authors cannot claim the autophagy is inhibited (and probably is true but a detailed study in autophagic function should be performed to draw that conclusion). The reviewer recommends to tone down that conclusion because, according to the findings in this study, it would be irrelevant given that the degradation of the mutant occurs through
Reviewer 2 Report
Comments and Suggestions for Authors
The authors revised this manuscript in accordance with the reviewer comments. The manuscript is well written. Fig.9B is not clear. The authors should move this figure to Supplemental figure.
Round 3
Reviewer 1 Report
Comments and Suggestions for Authors
As mentioned previously and indicted by the authors, the study focuses on the rapid GA-responsive degradation of the d270KD mutant, a protein that does not exist in nature. If the study does not provide any molecular clue about the (slow?) degradation of of wild-type CHK1, the significance of the paper is quite low.
The authors revised this manuscript in accordance with most of the reviewer comments. The manuscript is well written and the scientific soundness is OK. The research was conducted correctly and the paper has not serious flaws.However, the significance of the paper is quite low.
Although I clicked in the overall recommenadation " Reconsider after major revision (control missing in some experiments)", I have no recommendation. The Editorial Team should make the decision for publication (or not)
This reviewer feels that can not help to make the study better with any other comment. The Editorial Team must decide if the content of the manuscript is suitable or not for publication.